# Reliability of a Wearable Motion Tracking System for the Clinical Evaluation of a Dynamic Cervical Spine Function

**DOI:** 10.3390/s23031448

**Published:** 2023-01-28

**Authors:** Hamed Hani, Reid Souchereau, Anas Kachlan, Jonathan Dufour, Alexander Aurand, Prasath Mageswaran, Madison Hyer, William Marras

**Affiliations:** 1Spine Research Institute, The Ohio State University, Columbus, OH 43210, USA; 2Department of Integrated Systems Engineering, The Ohio State University, Columbus, OH 43210, USA; 3Center for Biostatistics, Department of Biomedical Informatics, The Ohio State University, Columbus, OH 43210, USA

**Keywords:** musculoskeletal disorder, neck function, reliability, intra-rater, inter-rater, wearables

## Abstract

Neck pain is a common cause of disability worldwide. Lack of objective tools to quantify an individual’s functional disability results in the widespread use of subjective assessments to measure the limitations in spine function and the response to interventions. This study assessed the reliability of the quantifying neck function using a wearable cervical motion tracking system. Three novice raters recorded the neck motion assessments on 20 volunteers using the device. Kinematic features from the signals in all three anatomical planes were extracted and used as inputs to repeated measures and mixed-effects regression models to calculate the intraclass correlation coefficients (ICCs). Cervical spine-specific kinematic features indicated good and excellent inter-rater and intra-rater reliability for the most part. For intra-rater reliability, the ICC values varied from 0.85 to 0.95, and for inter-rater reliability, they ranged from 0.7 to 0.89. Overall, velocity measures proved to be more reliable compared to other kinematic features. This technique is a trustworthy tool for evaluating neck function objectively. This study showed the potential for cervical spine-specific kinematic measurements to deliver repeatable and reliable metrics to evaluate clinical performance at any time points.

## 1. Introduction

Neck pain is the fourth most disabling condition worldwide with a point prevalence of 5.9% to 38.7% in adults. Not only is it prevalent, but it is on the rise [1]. Physiological, biomechanical, and psycho-social factors all contribute to the manifestation of neck pain. Because neck pain is a multifactorial condition, it can be challenging to pinpoint the root causes of the pain. Traditionally, patient surveys and questionnaires are used to quantify neck pain [2]. While these tools provide information regarding pain sensitivity and perception, They do not serve as accurate measures of neck pain and, thus, may not reflect the biomechanical functional limitations of the patient. Therefore, a quantitative functional measurement tool is desirable to provide reliable results to assess specific functional status associated with neck disorders.

One way to capture the functional status of the neck is to study the kinematics of its movement. Various measurement tools have been used to quantify neck kinematics. Some studies used more traditional tools such as dynamometers and goniometers to measure body kinematics [3,4,5]. Recent developments in wearable inertial measurement units (IMUs) have provided researchers with more accurate tools that can capture the complex biomechanical kinematics of human motion [6]. However, for such metrics to be meaningful, one must understand the reliability and repeatability of the results produced by the measurements. According to the Six Sigma process, it is a crucial step to analyze and limit the variance of a measurement system, so that variance is attributed to factors being measured by the tool as opposed to measurement system noise or error [7].

Different metrics have been used to assess tool reliability including intraclass correlations, standard error of measurement, Bland–Altman plots, Cohen’s kappa, and Pearson’s r, yet it is still debatable as to which method is most appropriate [8]. Maynard et al. [9] assessed the reliability of gait measures from an optical motion analysis system using the intraclass correlation coefficient (ICC) method and the Bland and Altman test, whereas Lee et al. [10] used Cohen kappa to report inter-rater reliability of inertial measurement units (IMUs) when used to measure the motion of different body parts. The advantage of using ICCs is that the reliability quantified through this method enables researchers to assess observer effects [11]. According to McGraw and Wong [12], there are 10 types of ICCs, which are used in different experimental settings. However, most research papers fail to report which method was used for the reported ICC values making interpretation and comparison of those papers challenging. Thus, it is crucial to establish the type of ICCs when reporting the reliability of a measurement system.

Previous studies have reported a range of reliability ICCs for IMU-based metrics. Yoon et al. [13] showed the reliability of IMUs in measuring cervical range of motion to be good to high in 33 healthy patients. Carmona-Pérez et al. [14] assessed the reliability and validity of IMUs for the assessment of craniocervical range of motion (ROM) in cerebral palsy patients and showed intra-day reliability of 0.82 to 0.93 when measuring range of motion in all three anatomical planes. Chalimourdas et al. [15] showed the reliability of an IMU-based device designed to measure the cervical range of motion using 36 healthy individuals. The results showed ICC values between 0.54 to 0.9 for a two-way, mixed effect, absolute agreement model. As described in the literature, most of the studies investigate the range of motion and ignore higher degrees of motion, speed, and acceleration, which contain valuable information about the intrinsic nature of functional status [16].

Despite the popularity of IMU-based platforms, there are some drawbacks in using them which make results challenging to interpret. Long-term drift, magnetic interference, and inconsistency are some of the most critical issues involved with IMU-based platforms. However, several methods have been developed to reduce potential issues, including the use of a three-axis magnetometer to measure the magnetic field direction in the proximity of sensors [17].

The goal of this study was to determine the degree to which the cervical spine’s kinematic properties, as measured by an integrated software/IMU-based platform, were reliable. To comprehend the total variation among the raters at various time points, inter- and intra-rater reliability must be assessed. Future research that uses the same or comparable systems will build on this study’s findings.

## 2. Materials and Methods

The methods in this study were for the most part identical to the previous study conducted by the same researchers [18].

### 2.1. Subjects

In this study, a total of 20 healthy participants—10 men and 10 women, ages 18 to 64—were assessed. Flyers and word-of-mouth were used to find participants. Given that prior studies had proven that 20 participants were necessary to determine statistical significance, this quantity was chosen [19].

#### 2.1.1. Inclusion Criteria

Participants in this research must be between 18 and 80 to be eligible. To grasp instructions and safety concerns, participants had to understand English and have the ability to stand for at least 20 minutes for each motion evaluation.

#### 2.1.2. Exclusion Criteria

Neck pain or a history of chronic neck pain lasting longer than three months, severe vision loss, hearing loss, recent musculoskeletal fractures (within three months), and recent radiation or chemotherapy therapy were among the exclusion criteria for subjects. The researchers exercised discretion for any additional unlisted medical condition that may have made doing the test in a safe manner impossible.

### 2.2. Clinical Neck Motion Assessment

#### Platform Components

Data Collection, filtering, reviewing, sorting, processing, and storage were performed in a custom cloud-based software application. Functional cervical spine motion evaluation was carried out using highly accessible IMU sensors (XSens MTw2) incorporated into unique, lightweight, and flexible harnesses. The wearable motion device was designed especially for the noninvasive gathering of cervical-specific kinematics. The sensors were installed on harnesses worn over clothing to allow for secure human interaction. Figure 1 shows the sensors’ configuration for cervical spine motion assessment. One of the harnesses was fixed on the head, placing the sensor in the middle of the forehead. The other sensor was placed in the middle of the shoulders and aligned with the spine on a harness similar to a backpack. To correctly adjust for sensor drift, a custom algorithm was used with all hardware and software parts when recording the kinematic variables [20].

### 2.3. Study Design

To evaluate intra- and inter-rater reliability, a repeated measure experimental design with three raters was employed. The researchers chose naive volunteers as their raters for motion evaluations. Prior to subject recruitment and data collection, participants received technology usage training as part of the study protocol. On three different days, the three raters individually assessed each participant (9 total data collection sessions). On the same day, each subject had a 30-minute break in between motion evaluations before each test. To lessen the learning effect and exhaustion on participants, 2 days of resting were also necessary after each test day. The order of the raters was counterbalanced among the participants and the days. Tests were administered by raters blinded to the other rater’s performance. Table 1 displays a typical participation schedule.

### 2.4. Testing Procedures

On a participant’s first visit, the subject was provided sufficient time to review the consent form and ask any additional questions. Lightweight harnesses (headband and vest) were placed on the subject to allow the mounting of sensors as shown in Figure 1. Subjects then performed a series of motion trials, performing neck motions in all of three main anatomical planes (flexing and extending, bending right to left, and twisting right to left), as well as combinations of these planes (Illustrated in Figure A1). For each trial, the program gave the participant verbal instructions and instructed them to go as quickly or as far as they could comfortably. Range of motion was acquired in the first three motion trials, whereas dynamic movements were collected in the final five trials. Figure A2 shows an example of collected signals. The subjects underwent 9 data-collecting sessions before completing the study (3 sessions per rater).

### 2.5. Statistical Analysis

Features, such as range, maximum, minimum of flexibility, velocity, and acceleration signals were among the parameters used to examine neck motion data. An additional section as the appendix contains an outline and description of these characteristics. To further examine the intra-rater and inter-rater reliability of cervical spine motion ratings, intraclass correlation coefficients (ICCs) were computed using repeated measures, mixed-effect regression models. Koo and Li [21] introduced a guideline to compute different types of ICCs and how to interpret them which ICC(2,1) and ICC(3,1) were chosen based on the design of this study. ICC(2,1) and ICC(3,1) are calculated using Equations (Equation 1) and (Equation 2), respectively.
(1)$$\mathrm{ICC}\left(2,1\right)=\frac{MS_{r}-MS_{e}}{MS_{r}+\left(k-1\right)MS_{e}+\frac{k}{n}\left(MS_{e}-MS_{c}\right)}$$
(2)$$\mathrm{ICC}\left(3,1\right)=\frac{MS_{r}-MS_{e}}{MS_{r}+\left(k-1\right)MS_{e}}$$
where $MS_{r}$ is between-target mean square, $MS_{c}$ is between-rater mean square in case of inter-rater and between-day mean square in case of intra-rater reliability, $MS_{e}$ is residual mean square traditionally referred to as mean square error calculated from two-way analysis of variance (ANOVA). *k* is the number of raters rating each target in case of the inter-rater and the number of days each target is measured in case of intra-rater reliability. *n* is the total number of targets.

According to this guideline ICC estimates below 0.5, from 0.5 to 0.75, from 0.75 to 0.9, and above 0.9 suggest poor, moderate, good, and excellent reliability, respectively. R and SAS 9.4 were used for all analyses. α = 0.05 represented the type I error rate.

## 3. Results

### 3.1. Demographics

The subjects’ demographic backgrounds are shown in Table 2; 70% (*n* = 14) of the population were Caucasian, 25% (*n* = 5) were Asian, and 5% (*n* = 1) were unspecified. Additionally, the mean values for the participants’ heights and weights were 171.5 ± 9.6 cm and 69.8 ± 11.9 kg, respectively.

### 3.2. Intra-Rater Reliability

A total of 37 kinematic features during functional motion assessments were extracted and examined for reliability. Detailed descriptions of each measure and its interpretation are presented in Table A1 and Table A2. For each motion task, the intra-rater ICC estimates and 95% confidence intervals are displayed as a function of the cardinal planes. The intra-rater reliability (shown in Table 3) for all motion metrics in the axial plane were excellent, with ICC values of 0.94 (95% CI: 0.90–0.97), 0.94 (95% CI: 0.89–1.0), and 0.92 (95% CI: 0.83–1.0) for axial flexibility, velocity, and acceleration measures, respectively.

In the lateral plane, all flexibility motion metrics also yielded excellent reliability estimates, with mean ICC = 0.94 (95% CI: 0.88–0.97). Furthermore, mean lateral velocity metrics also ranged from good to excellent, ICC = 0.9 (95% CI: 0.85–0.96). In addition, lateral acceleration metrics showed good reliability, with right lateral acceleration showing the highest reliability of ICC = 0.89 (95% CI: 0.81–1.0).

Lastly, we discovered excellent ICC estimates after examining all the intra-rater reliability values in the sagittal anatomical plane, with mean ICC estimates ranging between 0.90–0.95 for flexibility, velocity, and acceleration measures.

For all multi-planar motion metrics of flexibility, velocity, and acceleration measurements, the intra-rater ICC values show good to excellent reliability.

### 3.3. Inter-Rater Reliability

When compared to intra-rater reliability, the inter-rater reliability results (Table 4) showed a different trend. All motion metrics in the axial plane resulted in good inter-rater reliability, with mean values ranging from 0.79–0.87. Overall, the highest inter-rater reliability for the axial plane was for left axial velocity with ICC = 0.87 (95% CI: 0.77–1.0). With mean ICC estimates ranging from 0.78 to 0.85 in the lateral plane, flexibility and velocity measures yielded good inter-rater reliability estimates, and acceleration metrics showed moderate reliability estimates, with mean ICC estimates ranging from 0.70–0.75.

For the sagittal symmetric tasks, the most reliable metric was sagittal velocity with an ICC = 0.89 (95% CI: 0.82–1.0). The best metric for the sagittal asymmetric tasks was extension velocity (ICC = 0.86, 95% CI: 0.79–1.0). All other metrics for sagittal symmetric and asymmetric tasks produced good reliability.

### 3.4. Trend of Reliability of Metrics

An interesting trend in the ICC distributions is revealed when the inter-rater and intra-rater data are combined. For instance, in the axial and sagittal planes, velocity metrics exhibited greater ICCs for all the motion assessments relative to both flexibility and acceleration metrics, but flexibility metrics displayed the greatest ICC values in the lateral plane. Figure 2, Figure 3 and Figure 4 represent ICCs of the inter-rater vs. intra-rater reliability for all of the metrics in the axial, lateral, and sagittal planes, respectively.

In order to understand the relative relationship between the various kinematic measures, these plots were created to illustrate the estimations of the ICC distributions. In order to provide for a more comprehensive analysis of the correlations between the intra-rater and inter-rater ICC mean estimates, the reliability areas for poor, moderate, good, and excellent ratings are presented in these figures in red, yellow, light green, and dark green, respectively. Circles represent positional measurements, triangles represent velocity measurements, and squares represent acceleration measurements. According to these figures, the mean ICC values for all measures were good for the axial and sagittal planes and moderate to good for the lateral plane. These plots show two key patterns that may be identified. First, the estimates of ICC values for inter-rater and intra-rater reliability within the sagittal and axial planes of the body were mostly higher than the values for the lateral plane of the body. Second, ICC values for the velocity kinematic metrics were almost always higher compared to the other kinematic metrics in all planes of the body.

## 4. Discussion

This study shed light on the reliability of various cervical kinematic metrics from a wearable IMU-based cervical motion monitoring system. The link between spine kinematics and its use as a functional indicator has been established in the literature [22]. This study further highlights the importance of recording reliable kinematic data to determine the cervical spine status.

Optical motion capture systems are the gold standard for accurate 3D measurement of kinematics but require a big laboratory setup and are expensive to acquire, operate, and maintain. On the other hand, IMUs are accessible at a low price and can be used in custom setups and environments due to properties, such as lightweight and flexibility. The technique utilized here is incredibly accurate in measuring spine motion. IMUs were compared to a high-fidelity visual motion capture system and found to be within about 99% of the vision system’s accuracy [20]. Therefore, the reliability values can be attributed to the repeatability of the test as a result of the variance in raters’ performance and discrepancies over time.

In this study, inter-rater and intra-rater reliability of kinematics measures derived from various tasks showed moderate to excellent reliability. In general, intra-rater reliability ICCs were slightly higher than inter-rater reliability ICCs. It is important to note that the confidence intervals for these ICCs are large, so concluding that intra-rater ICCs are significantly larger than inter-rater ICCs is premature. It is also important to note that the intent of this effort was indeed to “expose” the motion features that have lower ICCs so they can be excluded from use in future algorithms. Many features have high intra- and inter-rater ICC that can be prioritized during future algorithm development while features with low intra- and inter-rater ICC can be avoided.

The main sources of variability truly coming from the raters would primarily be from the placement of the harnesses on the subject and any human discretion to recollect a motion in the case that the subject performed it incorrectly and the software did not flag it. The rest of the protocol is automated, and the subject receives primary instructions from the computer. Given that the software algorithms are intentionally designed to minimize the effect of harness placement (there is a wide range of “correct” harness locations), the slightly inflated variability that is driving this phenomenon is likely the result of the effect of the order being conflated with the effect of rater within a given day.

Another important observation was that velocity-related metrics proved to produce more reliable results compared to the other metrics. In axial and sagittal planes, velocity-related metrics were among the most reliable metrics, whereas in the lateral plane, flexibility metrics produced more reliable results. This information can prove to be valuable when developing models for cervical spine status quantification providing the most reliable and useful features.

Our ICCs are comparable to the same values reported by other studies investigating the cervical spine, and in many cases, are even better. Besides the studies mentioned in the Introduction section, other studies have tried to evaluate the reliability of IMUs to measure cervical motion. Fletcher and Bandy [23] investigated the reliability of measuring cervical active range of motion for 25 individuals with neck pain and 22 individuals without neck pain using a cervical range-of-motion (CROM) device. The results showed ICC(3,1) values from 0.87–0.94 for the subjects without neck pain and 0.88–0.96 for the subjects with neck pain. Anoro-Hervera et al. [24] assessed intra-rater and inter-rater reliability of IMU-measured cervical active range of movements in 20 young asymptomatic adults with two raters. They reported ICC(3,1) values above 0.9 for intra-rater reliability and ICC(3,2) values above 0.75 for inter-rater reliability.

Moreover, using the CROM device, Audette et al. [25] evaluated the test-retest reliability of the range of motion values between two testing days using ICC(3,3). In that study, flexion resulted in the lowest ICC value (ICC = 0.89, 95% CI: 0.73–0.96), and extension resulted in the highest ICC value (ICC = 0.98, 95% CI: 0.95–0.99). Stenneberg et al. [26] calculated inter-rater reliability using ICC(2,1) for two raters conducting the test on symptomatic patients using a smartphone application and a Polhemus Liberty. The study reported flexion/extension to have the lowest reliability (ICC = 0.90, 95% CI: 0.78–0.95) and rotation to have the highest reliability (ICC = 0.96, 95% CI: 0.90–0.98). Reliability measures have also been calculated for the OSI CA 6000 spine motion analyzer, as shown in the study by Petersen et al. [27]. This study evaluated the reliability of the cervical motion measurements of healthy and symptomatic participants. ICC(2,1) was used for intra-rater measures for both subject groups and ICC(2,k) was used for inter-rater measures for healthy subjects. In healthy subjects, the lowest inter-rater reliability measure was reported for extension (ICC = 0.88), and the highest inter-rater reliability measure was reported for right rotation (ICC = 0.94). The lowest intra-rater reliability measure for healthy subjects was seen in flexion (ICC = 0.78), whereas the highest intra-rater reliability for healthy subjects was seen in left-side bend and right rotation (ICC = 0.94). Finally, the lowest and highest intra-rater ICC values for symptomatic participants were seen in flexion (ICC = 0.68) and left-side bend (ICC = 0.96), respectively.

In general, our ICC values indicated that intra-rater reliability is better than inter-rater reliability. This suggests that there was somewhat more agreement between testing days (time points) than there was between raters. Additionally, rather than the raters or testing procedure, the participants’ daily variations may potentially be the cause of the measures’ variability. Given that the subjects did not experience any appreciable changes in stiffness or discomfort, it is plausible that uncontrollable additional factors such as sleep, any physical activity prior to participating in the study, nutrition, or psychological and social changes influenced the subject’s ability to move. It is also likely that the learning effect occurred in between sessions as a deeper analysis did indicate that the first test day’s intra- and inter-rater agreement of metrics was worse than the same metrics on the following test days.

There were a number of key points that can be considered limitations of this study. As previously noted, various external factors, such as recent intense athletic activities and psychological issues, which may have affected motion evaluations were not measured. Because motion evaluations given on the same day would probably not exhibit significant differences in these parameters, some of them may be eliminated or reduced by the study design and reliability computations. Furthermore, it was anticipated that repetition of the movements would make participants more familiar with the protocols. The other limitation of this study included only asymptomatic subjects, which hindered the expansion of the results of this study to subjects experiencing neck pain. Another possible drawback was that additional, complicated features besides the conventional kinematic measurements that may serve as reliable metrics were not investigated in the analysis of the motion signals.

Despite all of the limiting variables, there was relative consistency between and among raters, as shown by the results of good to excellent ICC values for measures with intra-rater reliability and moderate to excellent values for measures with inter-rater reliability.

## 5. Conclusions

Overall, ICCs proved a proficient level of inter-reliability and intra-rater reliability for the cervical wearable motion tracking system. The intra-rater reliability was better than the inter-rater reliability for each measure. The kinematic metrics’ intra-rater reliability ratings varied from good to excellent, with most metrics showing excellent reliability. The majority of the inter-rater reliability metrics showed good repeatability except for three metrics in the lateral plane. Except the lateral plane, measures extracted from the velocity signals in all anatomical planes showed greater reliability ratings than the flexibility metrics. Overall, the reliability results indicate that this kinematic cervical motion tracking system has a high reliability for testing neck motion for all the raters during a single day. The metrics also showed some slight day-to-day variation, which is most likely the effect of daily changes in the subject’s performance. This study showed that kinematic neck motion metrics, as assessed by our technology, may give excellent, reproducible metrics to evaluate an individual’s spine kinematic state across time regardless of the raters familiarity with the device. 

## Figures and Tables

**Figure 1 sensors-23-01448-f001:**
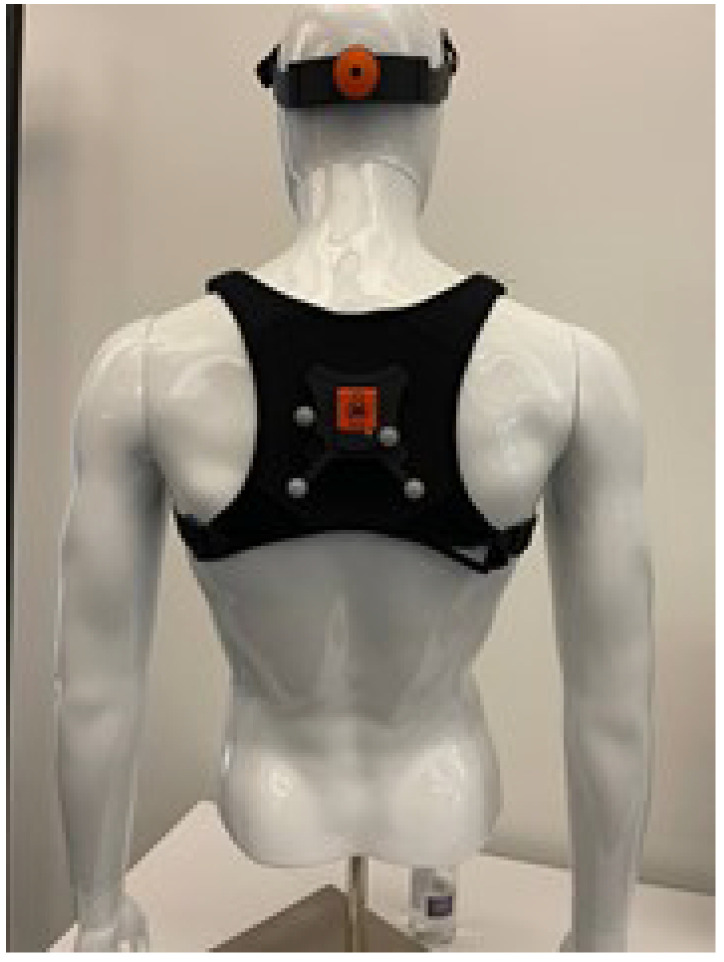
IMU sensors’ configuration for cervical spine motion assessment.

**Figure 2 sensors-23-01448-f002:**
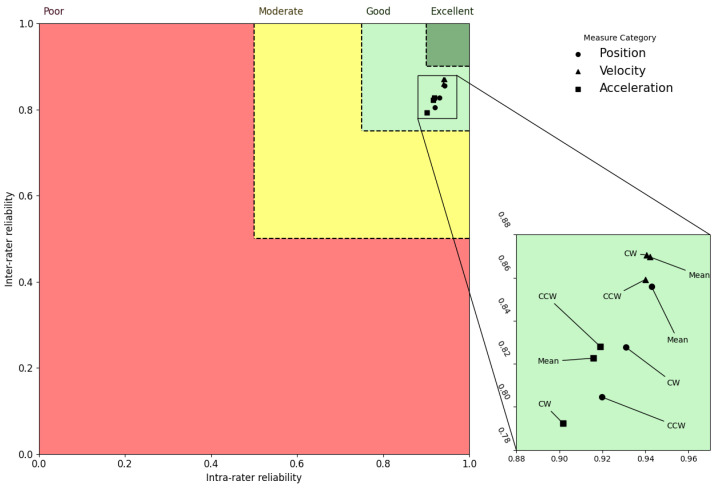
Axial plane inter-rater vs. intra-rater reliability. Regions with poor, moderate, good, and excellent reliability are shown, respectively, in red, yellow, light green, and dark green. Circles, triangles, and squares are used to represent measures extracted from flexibility, velocity, and acceleration signals, respectively.

**Figure 3 sensors-23-01448-f003:**
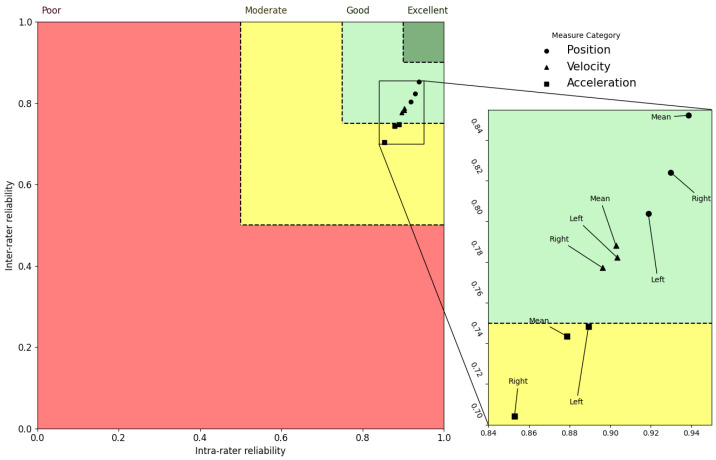
Lateral plane inter-rater vs. intra-rater reliability. Regions with poor, moderate, good, and excellent reliability are shown, respectively, in red, yellow, light green, and dark green. Circles, triangles, and squares are used to represent measures extracted from flexibility, velocity, and acceleration signals, respectively.

**Figure 4 sensors-23-01448-f004:**
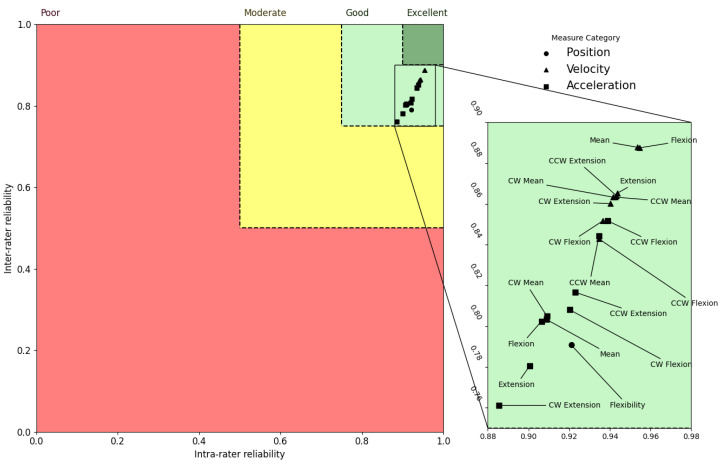
Sagittal plane inter-rater vs. intra-rater reliability. Regions with poor, moderate, good, and excellent reliability are shown, respectively, in red, yellow, light green, and dark green. Circles, triangles, and squares are used to represent measures extracted from flexibility, velocity, and acceleration signals, respectively.

**Table 1 sensors-23-01448-t001:** Randomization of 3 raters during 3 days of data collection each day (9 recordings).

Recording	1	2	3	4	5	6	7	8	9
Day	1			2			3		
Test	1	2	3	1	2	3	1	2	3
Rater	I	II	III	II	III	I	III	I	II

**Table 2 sensors-23-01448-t002:** Highlights of the subjects’ characteristics.

Characteristics	Mean (Std) or *n* (%)
Age	
18–20	1 (5%)
20–25	8 (40%)
25–30	2 (10%)
30–35	3 (15%)
35–40	1 (5%)
40–45	-
45–50	-
50–55	2 (10%)
55–60	1 (5%)
Over 60	2 (10%)
Gender	
Female	10 (50%)
Male	10 (50%)
Race	
Asian	5 (25%)
Caucasian	14 (70%)
Unspecified	1 (5%)
Height and Weight	
Height (cm)	171.5 (9.6)
Weight (kg)	69.8 (11.9)

**Table 3 sensors-23-01448-t003:** ICC values estimated for intra-rater reliability of all features.

Measure	ICC	95% Confidence Intervals	
		Interval Low (2.5%)	Interval High (97.5%)
CW Axial Flexibility	0.93	0.87	0.97
CCW Axial Flexibility	0.92	0.84	0.96
Mean Axial Flexibility	0.94	0.9	0.97
CW Axial Velocity	0.94	0.89	1
CCW Axial Velocity	0.94	0.89	1
Mean Axial Velocity	0.94	0.89	1
CW Axial Acceleration	0.9	0.79	1
CCW Axial Acceleration	0.92	0.83	1
Mean Axial Acceleration	0.92	0.82	1
Right Lateral Flexibility	0.93	0.87	0.96
Left Lateral Flexibility	0.92	0.84	0.96
Mean Lateral Flexibility	0.94	0.88	0.97
Right Lateral Velocity	0.9	0.84	0.94
Left Lateral Velocity	0.9	0.85	1
Mean Lateral Velocity	0.9	0.85	0.96
Right Lateral Acceleration	0.85	0.77	0.93
Left Lateral Acceleration	0.89	0.81	1
Mean Lateral Acceleration	0.88	0.8	1
Flexion Sagittal Flexibility	0.92	0.88	0.95
Flexion Sagittal Velocity	0.95	0.93	1
Extension Sagittal Velocity	0.94	0.91	1
Mean Sagittal Velocity	0.95	0.92	1
Flexion Sagittal Acceleration	0.91	0.85	1
Extension Sagittal Acceleration	0.9	0.85	1
Mean Sagittal Acceleration	0.91	0.86	1
CCW Sagittal Flexion Velocity	0.93	0.9	1
CCW Sagittal Extension Velocity	0.94	0.91	1
CCW Sagittal Mean Velocity	0.94	0.91	1
CCW Sagittal Flexion Acceleration	0.94	0.9	1
CCW Sagittal Extension Acceleration	0.92	0.88	1
CCW Sagittal Mean Acceleration	0.93	0.89	1
CW Sagittal Flexion Velocity	0.94	0.91	1
CW Sagittal Extension Velocity	0.94	0.91	1
CW Sagittal Mean Velocity	0.94	0.91	1
CW Sagittal Flexion Acceleration	0.92	0.87	1
CW Sagittal Extension Acceleration	0.89	0.82	1
CW Sagittal Mean Acceleration	0.91	0.85	1

**Table 4 sensors-23-01448-t004:** ICC values estimated for inter-rater reliability of all features.

Feature	ICC	95% Confidence Intervals	
		Interval Low (2.5%)	Interval High (97.5%)
CW Axial Flexibility	0.83	0.71	0.9
CCW Axial Flexibility	0.8	0.65	0.89
Mean Axial Flexibility	0.86	0.76	0.92
CW Axial Velocity	0.87	0.77	1
CCW Axial Velocity	0.86	0.76	1
Mean Axial Velocity	0.87	0.77	1
CW Axial Acceleration	0.79	0.6	1
CCW Axial Acceleration	0.83	0.66	1
Mean Axial Acceleration	0.82	0.64	1
Right Lateral Flexibility	0.82	0.7	0.89
Left Lateral Flexibility	0.8	0.66	0.88
Mean Lateral Flexibility	0.85	0.74	0.92
Right Lateral Velocity	0.78	0.68	0.87
Left Lateral Velocity	0.78	0.69	1
Mean Lateral Velocity	0.79	0.69	1
Right Lateral Acceleration	0.7	0.58	0.85
Left Lateral Acceleration	0.75	0.6	1
Mean Lateral Acceleration	0.74	0.6	1
Flexion Sagittal Flexibility	0.79	0.7	0.86
Flexion Sagittal Velocity	0.89	0.82	1
Extension Sagittal Velocity	0.87	0.79	1
Mean Sagittal Velocity	0.89	0.82	1
Flexion Sagittal Acceleration	0.8	0.69	1
Extension Sagittal Acceleration	0.78	0.68	1
Mean Sagittal Acceleration	0.8	0.7	1
CCW Sagittal Flexion Velocity	0.84	0.77	1
CCW Sagittal Extension Velocity	0.86	0.79	1
CCW Sagittal Mean Velocity	0.86	0.79	1
CCW Sagittal Flexion Acceleration	0.85	0.77	1
CCW Sagittal Extension Acceleration	0.82	0.73	1
CCW Sagittal Mean Acceleration	0.84	0.76	1
CW Sagittal Flexion Velocity	0.85	0.79	1
CW Sagittal Extension Velocity	0.86	0.78	1
CW Sagittal Mean Velocity	0.86	0.8	1
CW Sagittal Flexion Acceleration	0.81	0.71	1
CW Sagittal Extension Acceleration	0.76	0.65	1
CW Sagittal Mean Acceleration	0.8	0.7	1

## Data Availability

This study analyzed patient health information, which cannot be publicly available.

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
