# Peer review of "Reliability of a Wearable Motion Tracking System for the Clinical Evaluation of a Dynamic Cervical Spine Function"

_sensors, 2023, doi:10.3390/s23031448_

Round 1

Reviewer 1 Report

In this manuscript, the authors evaluated the reliability of wearable IMU for the spine kinematic state assessment with ICC. This study is interesting for related researchers, and following are my specific comments:

1.    The calculation of ICC is the fundamental element of this study. Although the authors provide some reference for ICC calculation, it would be better if the authors can describe the ICC calculation in the context of IMU data in the manuscript.

2.    More details about the sensors and their arrangement should be provided.

3.    A comparison between the IMU-based evaluation and other methods should be included in the manuscript to show the advantage of the IMU-based method.

Author Response

Response to Reviewer 1 Comments

  1. The calculation of ICC is the fundamental element of this study. Although the authors provide some reference for ICC calculation, it would be better if the authors can describe the ICC calculation in the context of IMU data in the manuscript.

Response 1: Formulation and description of ICC calculation was added to “2. Materials and Methods” section under “2.5 Statistical Analysis” subsection.

  1. More details about the sensors and their arrangement should be provided.

Response 2: In “2. Materials and Methods” section under subsection “Platform components” (line 97) a brief descriptoin of sensors is provided in the revised manuscript. The model of used sensors is XSens MTw 2 (https://www.xsens.com/products/mtw-awinda) which is a well-known model in the ergonomics field. Details of the sensor alignement and configuration are also described in the same paragraph and shown in Figure 1.

  1. A comparison between the IMU-based evaluation and other methods should be included in the manuscript to show the advantage of the IMU-based method.

Response 3: In line 210, the paragraph was editted to show the benefit of using IMUs.

“Motion capture system is the gold standard for position, velocity, and acceleration measurement that requires a big laboratory setup and is expensive to acquire, operate, and maintain. On the other hand, IMUs are accessible at a low price and can be used in custom setups and environments due to properties like lightweight and flexibility. The technique utilized here is incredibly accurate in measuring spine motion. IMUs were compared to a high-fidelity visual motion capture system and found to be within about 99% of the vision system's accuracy. Therefore, the reliability values can be attributed to the repeatability of the test as a result of the variance in raters' performance and discrepancies over time.”

Reviewer 2 Report

The manuscript reports on the reliability of quantifying neck function using a wearable cervical motion tracking system. Kinematic features from the signals are extracted and used as inputs to calculate the intraclass correlation coefficients (ICCs). The investigation performed on the study and ICCs estimation are interesting. And also, the manuscript is well organized. I recommend this manuscript for publication in this journal in its current form.

Author Response

Thanks for your feedback!

Reviewer 3 Report

The caption for Figure 1 seems to be misapplied, with the figure showing an example of subject instrumentation method.

In line 116, it should say "Tests were administered" or similar.

The caption for Table 3 contains a typo for "Feature"

Line 213 seems to be garbled in some way, consider rewording.

Perhaps the details of the method are contained in the referenced work by the same researchers (18), but what is the process for rater evaluation? Meaning, what do the researchers feel accounts for inter-rater variability being higher than intra-rater? Is there any subjective role for the individual raters? Or is the inter-rater differences due to use and setup of the IMUs on the subject only? A small amount of detail or discussion might help readers understand this.

Author Response

Response to Reviewer 3 Comments

  1. The caption for Figure 1 seems to be misapplied, with the figure showing an example of subject instrumentation method.

Response 1: Caption is fixed in the revised manuscript.

  1. In line 116, it should say "Tests were administered" or similar.

Response 2: This is fixed in the revised manuscript.

  1. The caption for Table 3 contains a typo for "Feature"

Response 3: Typo is fixed in the revised manuscript.

  1. Line 213 seems to be garbled in some way, consider rewording.

Response 4: This sentence (line 243 in the revised manuscript) is changed to: This information can prove to be valuable when developing models for cervical spine status quantification providing the most reliable and useful features.

  1. Perhaps the details of the method are contained in the referenced work by the same researchers (18), but what is the process for rater evaluation? Meaning, what do the researchers feel accounts for inter-rater variability being higher than intra-rater?

Response 5: The formulation for calculating ICC is added to the revised manuscript in “2. Materials and Methods” section under “2.5 Statistical Analysis” subsection which clarifies the difference between inter- and intra-rater reliability. Also, a paragraph in the “Discussion” section line 220 was added to clarify this point:

In general, we found that intra-rater reliability ICCs were slightly higher than inter-rater reliability ICCs. It is important to note that the confidence intervals for these ICCs are large, so we cannot conclude that intra-rater ICCs truly are significantly larger than inter-rater ICCs. It is also important to note that the intent of this effort was indeed to "expose" the motion features that have lower ICCs so they can be excluded from use in future algorithms. We found many features that have high intra-rater and inter-rater ICCs, so we will prioritize those in future algorithm development and avoid features with low inter-rater reliability ICCs.

  1. Is there any subjective role for the individual raters? Or is the inter-rater differences due to use and setup of the IMUs on the subject only? A small amount of detail or discussion might help readers understand this.

Response 6:  A paragraph in the “Discussion” section line 231 was added for further explanation.

The main sources of variability truly coming from the raters would primarily be from placement of the harnesses on the subject and any human discretion to recollect a motion in the case that the subject performed it incorrectly and the software did not flag it (which is very rare).  The rest of the protocol is automated and the subject receives primary instructions from the computer. Given that the software algorithms are intentionally designed to minimize the effect of harness placement (there is a wide range of "correct" harness locations), it is likely that the slightly inflated variability coming from the minor order effect is driving this phenomenon.